# Impact of vaccination against Japanese encephalitis in endemic countries

**G. William Letson**[1]*, **Anthony A. Marfin**[1], **Jessica Mooney**[1], **Huong Vu Minh**[2], **Susan L. Hills**[3], **the JE Vaccine Global Impact Assessment Team**[¶]

**1** PATH, Seattle, Washington, United States of America, **2** PATH Hanoi Towers, Hanoi, Vietnam, **3** Centers for Disease Control and Prevention, Division of Vector-Borne Diseases, Fort Collins, Colorado, United States of America

¶ Membership of the JE Vaccine Global Impact Assessment Team is provided in S1 Acknowledgements
* bwletson@gmail.com

**Data Availability Statement:** WHO/PATH survey data in excel files. Data presented in WHO/PATH BiRegional meeting power point files cannot be shared due to inability to remove some information

## Abstract

### Background

Japanese encephalitis (JE) virus is the leading cause of vaccine-preventable encephalitis and a significant cause of disability in Asia and the western Pacific. Many countries have introduced JE vaccination programs, including several low resource countries following WHO's prioritization of JE vaccination in 2006. We sought to characterize the public health impact of JE vaccination programs.

### Methodology/Principal findings

JE case data and vaccination coverage rates, were requested from country health officials in 23 JE endemic countries and Chinese Taipei. Additional data were extracted from meeting presentations and published literature. JE incidence was compared before and after vaccination using a minimum three year period pre and post program introduction or expansion. Data suitable for analysis were available for 13 JE-endemic countries and Chinese Taipei, for either all age groups or for children aged under 15 years only. Five countries and Chinese Taipei introduced vaccine prior to 2006 and the all-age JE incidence was reduced by 73–100% in about 5–20 years following introduction. Six countries have introduced JE vaccine since 2006, and JE incidence in children aged younger than 15 years has been reduced by 14–79% as of 2015–2021. JE-specific data were unavailable before introduction in Thailand and Vietnam, but vaccination programs reduced acute encephalitis incidence by 80% and 74%, respectively. Even in the programs with greatest impact, it took several years to achieve their results.

### Conclusions/Significance

JE vaccination has greatly reduced JE in 13 JE-endemic countries and Chinese Taipei. Highest impact has been observed in countries that introduced prior to 2006, but it often took roughly two decades and substantial resources to achieve that level of success. For greatest possible impact, more recently introducing countries and funding agencies should

that could be confidential and/or identifying of persons. However specific data without any identifying or confidential information from the power point presentations can be requested by cviadatarequests@path.org. Published data and data available in the public domain are cited in the manuscript or S1 and S2 Appendices.

**Funding:** Funding to PATH Japanese encephalitis Vaccine Introduction and Sustainability Program by the Bill and Melinda Gates Foundation (http://www.gatesfoundation.org) supported the study (Award Title: Ensuring sustainable supply and use of Japanese encephalitis vaccine; Award #OPP1115522 / AWD-252768). The funders had no role in study design, data collection and analysis, decision to publish, or preparation of the manuscript. Portions of salary for GWL, AAM, JM and HMV were paid by this grant.

**Competing interests:** The authors have declared that no competing interests exist.

commit to continuous improvements in delivery systems to sustain coverage after initial vaccine introduction.

## Author summary

While JE has been controlled by vaccination in a handful of economically developed countries endemic for JE transmission with disease reductions of 73–100%, introduction of JE vaccines has ocurred later in low and low-middle income countries (LLMIC) endemic for JE. Since 2006 focused effort has been undertaken to assist with establishment of sustained vaccination efforts against JE in LLMIC. The outcome has been growing evidence of the impact of vaccination against JE in LLMIC endemic for JE transmission with disease reductions of 14–79% among children aged under 15 years. We describe the public health impact of JE vaccination programs in both economically developed and LLMIC JE endemic countries and discuss how time and sustained effort on JE vaccination in LLMIC has high potential to reach the level of control seen in earlier decades in economically developed countries.

## Introduction

Japanese encephalitis (JE) virus is the leading cause of vaccine-preventable encephalitis and a significant cause of disability in Asia and the western Pacific [1]. Although JE virus can infect people of any age, in endemic areas, it is primarily a disease of children [2]. The case fatality ratio for symptomatic JE virus infection can be as high as 30%, while up to 50% of those who survive have significant long-term neurological or psychiatric sequelae [3]. In addition, symptomatic JE virus infections are associated with immense social and financial burden [4]. The most effective and sustainable way to prevent human JE virus infections is childhood vaccination [5].

Inactivated, mouse brain-derived (MBD) JE vaccines first became available in the 1930s. Initially, these vaccines were used in military personnel, and in the late 1960s, some JE-endemic Asian countries began wider introduction of these vaccines to national childhood vaccination programs. Since the expanded use of MBD JE vaccine in children, several newer vaccines have entered the market (e.g., live attenuated SA14-14-2 vaccine, inactivated Vero-cell derived vaccines, live chimeric vaccine). The live, attenuated JE vaccine, SA 14-14-2, is now the primary childhood JE vaccine used throughout Asia and the western Pacific [1].

An estimated 68,000 clinical cases of JE occur each year, with approximately 13,600 to 20,400 deaths [6]. In 2006, the World Health Organization (WHO) acknowledged JE vaccination as a public health priority, leading to guidelines on acute encephalitis syndrome (AES)/JE surveillance and JE vaccine introduction in hopes of expanding JE vaccination beyond the reatively wealthy countries that had already adopted JE vaccination policies [7]. Since 2003, through Bill and Melinda Gates Foundation funding, PATH has worked with WHO, UNICEF, Gavi, US Centers for Disease Control and Prevention, Chengdu Institute of Biological Products and others, in partnership with country ministries, agencies, and departments of health, to assist with the introduction and expansion of JE vaccination programs to all JE-endemic countries. Although several researchers have estimated the impact of vaccination using mathematical modeling [8–10], the impact of immunization has not been updated with actual surveillance data.

Measuring the impact of childhood vaccination on morbidity and mortality is critical to inform public health investment decisions made by national governments, WHO, Gavi, and other global donors. In 2021, the Vaccine Impact Modelling Consortium (VIMC), a consortium of infectious disease modelling institutions, modeled impact of vaccination against diseases caused by ten different pathogens, and estimated 100,000 deaths would be averted by JE vaccination from 2021 through 2030 [11]. Such mathematical models are heavily dependent on estimates of vaccine coverage and other measures of vaccine use over time. One of the stated limitations of the VIMC analysis was the uncertainty of estimates of past and future vaccine use, uptake, and coverage. Unlike the data regarding vaccine uptake, total vaccine doses used over time, and vaccine coverage for vaccines against global pathogens such as measles or polio, such data for recently introduced vaccines against regionally specific pathogens (e.g., JE, Group A meningococcus, typhoid) are less available and may be more variable in quality and content, especially in low-income and lower middle-income countries (LICs and LMICs).

The goal of our study was to characterize the public health impact of JE vaccine and to provide JE incidence and JE vaccination coverage data using actual disease surveillance data and to provide actual data to improve the utility of mathematical models that may inform future introductions or expansions of JE vaccination.

## Methods

Three different data collection methods were used to collect JE case counts and vaccination coverage data in 23 JE-endemic countries and Chinese Taipei in the Western Pacific (WPRO) and Southeastern Asia (SEARO) Regions of WHO. Data were not collected from Russia (WHO EURO region) or Pakistan (EMRO). Data collection methods were:

- *WHO-PATH protocol Survey Data*: A survey of the ministries, agencies, or departments of health in JE-endemic countries;

- *Biregional Data*: Extraction of case counts and incidence data from presentations made by ministries, agencies, and departments of health or their representatives at seven WPRO--SEARO Biregional JE meetings; and/or

- *Published Data*: Extraction of case counts and incidence data from reports published in epidemiological literature.

*WHO-PATH protocol Survey Data*. WHO regional and WHO country immunization officers and PATH staff developed a protocol and survey to collect JE and AES case counts and JE vaccination coverage data from national disease surveillance and national immunization program (NIP) directors in 23 JE-endemic countries and Chinese Taipei. The following data were collected for 2006–2018 when available:

- Number and type of JE vaccine doses given to children aged under 15 years in mass vaccination campaigns or routine immunization. To determine vaccine coverage, a child who received at least one dose of JE vaccine in a mass campaign or in routine immunization was considered fully vaccinated. This definition was used because it is consistent with WHO's single dose recommendation for use of the live attenuated SA14-14-2 JE vaccine [7], the primary vaccine used by most countries responding to the WHO-PATH survey, while recognizing the manufacturer and some country policies require a second dose;

- Number] or acute encephalitis [AE] cases based on local case definitions, in all persons and among children aged under 15 years; and

- Number of laboratory-confirmed (anti-JE IgM in CSF) and probable (anti-JE IgM in serum) JE cases in all persons and among children aged under 15 years.

*Biregional Data.* For countries that did not submit complete survey data, country-specific incidence data and vaccine coverage data were collected from the presentations at seven SEARO-, WPRO- and PATH-sponsored biregional meetings on the control of JE conducted between 2002–2016 [12–17]. At these meetings, JE and AES/AE surveillance data and JE vaccination coverage data were presented regularly by the same ministries, agencies, and departments of health who received the surveys noted above. Case definitions used in these presentations were consistent with those used for data provided in the survey.

*Published Data.* Finally, the Peoples Republic of China, the Republic of Korea, Japan, and Chinese Taipei began childhood JE vaccination programs 30 to 50 years ago and published results that were used to calculate impact for this study in these countries [18–23]. Additionally, available published data were used to refine, clarify, and corroborate data obtained through surveys or biregional meetings.

From these data, JE incidence before and after vaccine introduction was calculated using data for children aged under 15 years wherever available from the three data sources, or all ages data when that is all that was available. Only countries with JE surveillance data for at least three years before and three years after JE vaccine introduction or significant JE vaccination program expansion were included in this analysis. Because surveillance in some countries was established shortly before JE vaccine introduction, the pre vaccination period included the year of introduction when introduction was done late in the year. Some countries introduced JE vaccine initially in small subnational or regional populations. In these circumstances the before/after incidence calculation compared the transition from small regional to large regional or national introduction. Complete information on pre- and post introduction study periods and immunization program details for each country is provided in S1 and S2 Appendices. Because most of the reported JE incidence data are based on the serologic results of serum IgM ELISA, [24] the impact results reported here are considered to be the reduction of probable JE cases. Because CSF is infrequently obtained in evaluation of AES or AE, countries treat serum IgM as confirmatory evidence of JE in their surveillance data. Incidence rate per 100,000 was calculated by averaging the number of children aged under 15 years and/or all-age cases per year for the study periods before and after vaccine introduction and dividing by the average of children aged under 15 years and/or all-age population for these study periods [25], multiplied by 100,000.

For India, Indonesia, and Malaysia, data from one state or province were used to analyze impact. The data from Uttar Pradesh (India) represent impact from a large subnational JE vaccine introduction in a state which reported a high proportion (~75%) of all India's nationwide JE cases immediately prior to vaccine introduction; from Bali (Indonesia), data represent impact from vaccine introduction in the only province with a vaccination program; and from Sarawak (Malaysia), data represent impact from introduction primarily in Sarawak state where JE vaccination was targeted to all children aged under 15 years. We also had all ages data to assess JE reduction in Maylasia nationwide where outbreak targeted JE vaccination has been undertaken in provinces outside of Sarawak.

For countries with early JE vaccine introduction, all-age JE incidence was calculated until the first 4-year period where a 90% or greater reduction in incidence was identified (Fig 1). The number of years from the time of JE vaccine introduction to the first year of a 4-year period with sustained reduction was noted. A presumed ideal of 90% reduction is used here based on these countries having achieved a minimum reduction of 90% over time, demonstrating what is possible with long-standing and sustained efforts at national JE vaccination.

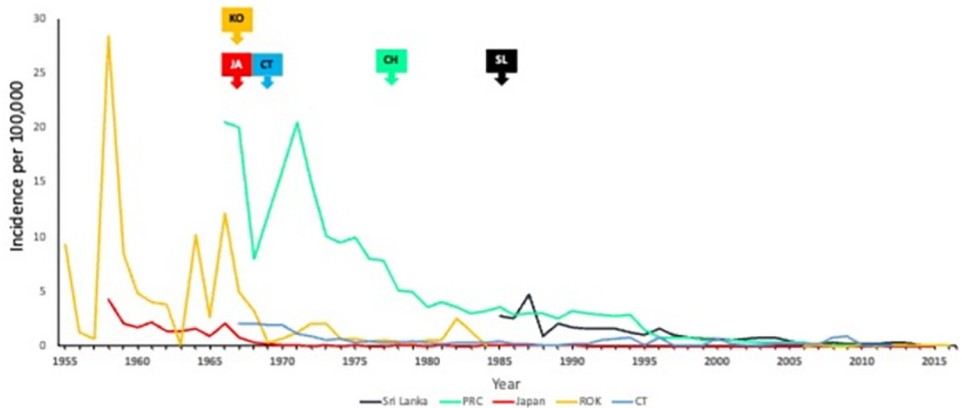

**Fig 1. All-age JE incidence in Sri Lanka (SL), People's Republic of China (CH), Japan (JA), Chinese Taipei (CT), and the Republic of Korea (KO) by year.** JE vaccine introduction noted by arrow.

For Thailand and Vietnam, national surveillance data prior to vaccine introduction were only available for AE cases and viral encephalitis cases, respectively. Therefore, the impact assessment for these countries was based on the reduction in reported AE or viral encephalitis cases. The etiology of encephalitis in up to 52% of encephalitis has been shown to be due to JE in these countries [26], and given samples are not always collected at optimal times, a higher percentage is likely to be due to JE. This suggests the impact on AE and viral encephalitis following vaccine introduction can likely be used as a surrogate measure for the impact of JE vaccine. We present post vaccine introduction laboratory-confirmed JE surveillance data in children aged under 15 years, available from the WHO-PATH survey, to validate use of AE/viral encephalitis reduction in these countries as a marker of JE reduction.

**Ethical Review:** As study did not use personal identifying information and was analysis of aggregate data it was declared exempt from review by the Ethics Review Committees of the Western Pacific Region office (WPRO) and the South East Asia Region office (SEARO) of the World Health Organisation.

## Results

Of the 23 JE-endemic countries and Chinese Taipei, 14 entities (13 countries and Chinese Taipei) were included in the impact analysis. Analyses were primarily based on survey data in seven cases, biregional data in three cases, and published data in four cases, although a variety of sources were typically considered. Nine countries were excluded from the vaccine impact analysis because they did not have a national JE vaccination program (Bangladesh, Bhutan, Papua New Guinea, Singapore, Timor Leste), had a program that was not sustained (Brunei, Democratic People's Republic of Korea), had a very localized program (Australia), or did not have a sufficiently long period of disease surveillance following introduction that allowed measurement of impact (Philippines). Disease surveillance type, summary of vaccine delivery over time, pre- and post-introduction study periods for analysis, vaccine coverage estimates, and data sources for the 14 entities that met analysis criteria are summarized in S1 and S2 Appendices.

### JE incidence before and after vaccine introduction or expansion

**JE Incidence in children aged under 15 years.**   Of six countries that introduced JE vaccine in or after 2006, four showed a decrease in JE incidence in children aged under 15 years of

**Table 1. JE incidence in children aged under 15 years before and after JE vaccine introduction, by country[1].**

| Country[2] (First year or vaccine introduction) | Incidence/100,000 children aged under 15-years-old | | Vaccine coverage | | Incidence change |
|---|---|---|---|---|---|
| | Pre-introduction (Study years) | Post-introduction (Study years) | Campaign (%) | Routine immunization (%) | |
| JE vaccine introduction before 2006 | | | | | |
| Republic of Korea (1967) | 11.9 (1960–1966) | 0.04 (2006–2018) | Not used for initial introduction | 99 | ↓100% |
| Malaysia (Sarawak) (2001) | 4.03 (1996–2001) | 0.68 (2006–2015) | None performed | 96 | ↓83% |
| JE vaccine introduction in 2006 or later | | | | | |
| Cambodia (2009) | 0.96 (2007–2015) | 0.20 (2016–2018) | 102 | 59 | ↓79% |
| Myanmar (2017) | 1.54 (2015–2017) | 0.62 (2018–2020) | 93 | 87 | ↓60% |
| Nepal (2006) | 1.57 (2006–2011) | 0.67 (2012–2018) | 106 | 71 | ↓57% |
| Indonesia (Bali) (2018) | 1.42 (2014–2017) | 0.64 (2018–2021) | 94 | 64 | ↓55% |
| Laos (2013) | 0.81 (2010–2014) | 0.62 (2015–2018) | 95 | 55 | ↓24% |
| India (Uttar Pradesh) (2006) | 5.50 (2005–2011) | 4.71 (2012–2018) | 96 | 50 | ↓14% |

[1] Additional information for each location on rationale for selection of pre- and post-introduction study periods, vaccine coverage estimates, and data sources are provided in S1 and S2 Appendices.

[2] For Malaysia, Indonesia, and India large subnational JE vaccine introductions have primarily occurred in the most affected state or province (shown in parentheses) and analysis reflects these large subregions.

50% or greater (Table 1). Two other countries showed more modest reductions of approximately 14% and 24%. The Republic of Korea and Malaysia both introduced vaccine prior to 2006 and showed a decrease in JE incidence in children aged under 15 years of 100% and 83%, respectively.

**All-age JE incidence.** Of five countries and Chinese Taipei that introduced JE vaccine before 2006, five showed a decrease in all-age JE incidence of 95% or greater (Table 2). Malaysia showed a decrease in all-age JE incidence of 73% throughout Malaysia and a reduction from 1.38 /100,000to 0.22/100,000 (84%) for all ages in Sarawak alone.

## Countries that obtained 90% reduction in JE incidence following vaccine introduction

Japan, the People's Republic of China, the Republic of Korea, Sri Lanka, and Chinese Taipei have achieved a 90% or higher reduction in JE incidence following national introduction of JE vaccine (Table 2 and Fig 1). Although Japan achieved 90% reduction in the first 4-year period immediately following vaccine introduction, it took several years for the other countries to achieve sustained 90% reduction (Republic of Korea– 6 years; Chinese Taipei– 16 years; People's Republic of China– 18 years; Sri Lanka– 18 years). This is likely due to the time needed to identify the children at greatest risk for disease and to develop a JE vaccine delivery system to adequately cover those at-risk populations (S2 Appendix). We did not examine this further in our study.

**Table 2. JE incidence in persons of any age before and after JE vaccine introduction, by country or autonomous region[1].**

| Country[2] (Vaccine introduction) | Incidence/100,000 population | | Vaccine coverage | | Incidence change |
|---|---|---|---|---|---|
| | Pre-introduction (Study years) | Post-introduction (Study years) | Campaign (%) | Routine immunization (%) | |
| JE vaccine introduction before 2006 | | | | | |
| Republic of Korea (1967) | 5.5 (1960–1966) | 0.03 (2006–2018) | Not used for initial introduction | 99 | ↓99% |
| Malaysia- entire country, JE vaccine mostly in Sarawak (2001) | 2.6 (1996–2001) | 0.7 (2006–2015) | None performed | 96 | ↓73% |
| People's Republic of China (1980) | 17.0 (1967–1970) | 0.2 (2010–2013) | Not used for introduction | 95 | ↓99% |
| Chinese Taipei (1968) | 1.9 (1966–1968) | 0.1 (2008–2012) | None performed | ~90–95 | ↓95% |
| Sri Lanka (1988) | 2.8 (1985–1987) | 0.14 (2013–2015) | 95 | 90 | ↓95% |
| Japan (1967) | 2.7 (1950–1967) | 0.004 (1991–2015) | None performed | 83–100[3] | ↓100% |

[1]Additional information for each location on rationale for selection of pre- and post-introduction study periods, vaccine coverage estimates, and data sources are provided in S1 and S2 Appendices.

[2]For Malaysia, a large subnational JE vaccine introduction has only occurred in Sarawak, the most affected state.

[3] In period from 2005–2008 when recommendation for vaccination was suspended, coverage was only 4%

### AE or viral encephalitis incidence in countries without laboratory-confirmed JE surveillance data prior to JE vaccine introduction (Thailand and Vietnam)

In Thailand, all-age AE incidence was measured before and after vaccine introduction. For the period 1977–1989, the AE incidence rate was 4.4 cases per 100,000. In 2011–2015 (the most recent 5-year period with data available), after introduction of vaccine in 1990, the all-age AE incidence was reduced to 0.9 cases per 100,000, representing an 80% reduction in disease. The lowest AE incidence during the full post-vaccination period was ~ 0.5/100,000 in 2003–2004. For 2011–2015, the JE incidence in children aged under 15 years was 0.2 per 100,000.

In Vietnam, all-age viral encephalitis incidence was measured before and after JE vaccine introduction [17]. For the period 1991–1996, the incidence was 3.8 cases per 100,000. In 2015–18, after introduction of vaccine in 1997 and full national expansion by 2014, the all-age viral encephalitis incidence was reduced to 1.0 cases per 100,000, representing a 74% reduction in viral encephalitis. AE incidence gradually decreased from 1997 onwards, reaching its lowest level during 2015–18. For 2009–2018, the JE incidence in children aged under 15 years was 0.9 per 100,000.

### Discussion

JE vaccination has greatly reduced JE, AES, AE and viral encephalitis in 13 JE-endemic countries and Chinese Taipei in Asia and the Western Pacific, but the impact has been highly variable. Most countries that introduced or expanded their vaccination programs nationally prior to 2006 reduced JE incidence by at least 90% while the reduction is less in countries that introduced in or after 2006 when WHO recognized JE vaccination in endemic countries as a public health priority. Because JE virus is a zoonotic virus, increasing impact over time is not due to

herd immunity as seen with many human-to-human transmitted viruses. Instead, it is likely due to improving surveillance to identify children at risk for JE and developing vaccine delivery systems that, over time, result in a greater proportion of at-risk children being individually protected against infection. Substantial social and ecological changes have taken place during recent decades and might have had a role in JE reduction. However, in countries without JE vaccination programs, such as Bhutan, JE virus transmission has persisted and in several other locations including some areas of Bhutan, Nepal, India, and Australia the geographical range and intensity of JE virus transmission recently has increased [6,27–30]. This suggests that social and ecologic factors alone are not responsible for the substantial reduction in JE incidence demonstrated in this assessment.

Most of the countries with greater impact are middle- and high-income countries and several had the highest JE burden in the region prior to the use of JE vaccine [1]. In these countries, JE has been nearly eliminated and has often become a disease of nonimmune adults rather than children [19–22,31,32]. There are many potential reasons for the high impact of JE vaccines including the development of highly efficient vaccination programs, intensive vaccination in rural areas where JE has its highest incidence, and other adjunctive JE control programs (e.g., vector control, husbandry practices). Perhaps most importantly, many of these countries had a JE disease burden and financial capacity to justify intensive JE control many years ago while other countries with lower burden or often less ability to identify JE disease burden were still focusing on improving JE surveillance and developing serologic testing capacity.

Although countries introducing JE vaccine on or after 2006 primarily used live attenuated SA14-14-2 JE vaccine which was not generally available for early-introducing countries, vaccine efficacy is unlikely to be a reason for the difference in impact. Countries such as Japan, Thailand, and the Republic of Korea, had very high vaccine impact using inactivated MBD JE vaccines. These inactivated MBD vaccines had an efficacy of 91% [33], but data reported from several studies have shown vaccine effectiveness of SA14-14-2, the most widely used JE vaccine since 2006, to be between 80% and 99% following single-dose vaccination and 98% or greater with 2 doses of the vaccine [34]. In addition, immunogenicity studies have demonstrated high seroprotection rates, and long-term vaccine effectiveness has been shown, following administration of a single dose of SA14-14-2 vaccine [35,36].

These data show that although still greater impact can be achieved in countries introducing JE vaccine on or after 2006, the progress made by those countries has been remarkable. With the assistance of the Bill & Melinda Gates Foundation, WHO, Gavi, and PATH, these countries have achieved roughly 15% to 80% reduction in JE incidence. Review of data in S1 and S2 Appendices reveals that the early adopting countries achieved relatively early and sustained routine immmunizaion JE vaccine coverage of > 85% and most of them > 90%. While countries adopting JE vaccination after 2006 achieved 85–90% coverage in supplementary immunization activities (SIA), the routine immunization coverage rose to a range of 50–71% for most, defining a possible explanation for reduced impact compared to early adopters. Most of these countries are LICs and LMICs where identifying at-risk populations and vaccine delivery in rural settings can be challenging. JE is primarily a rural disease, making vaccine uptake in rural populations critical for disease control. Yet, for newly established vaccination programs, vaccinating children in urban and peri-urban settings may be more readily accomplished. This could result in moderately high overall vaccine coverage for a country and still have relatively low coverage in the higher-risk rural areas. Where there is low JE vaccine coverage in rural areas, the impact of a program with otherwise reasonable coverage due largely to success in urban vaccination will be a lower than might be expected based on total coverage.

There are vaccine delivery practices in these later-introducing countries that could result in lower effectiveness and thus lower impact. The greatest concern is improper handling that reduces the potency of vaccine [37]. This may include the quality of the cold chain at all points of storage and at the site of administration, prolonged time between mixing lyophilized vaccines and administration, and over dilution of vaccine. In addition, practices that result in an overestimate of actual vaccine coverage could result in lower impact than expected. This would include factors such as multiple vaccine doses being given to some children while others (especially those at increased risk) do not receive any vaccine or the use of administrative records to estimate vaccine coverage [38]. Finally, one should consider whether, due to variable quality of JE serologic diagnostics in the pre-introduction study periods or the high prevalence of anti-JE IgM due to earlier asymptomatic infections among persons who then develop AES or viral encephalitis due to other causes, cases of childhood encephalitis that were labeled JE prior to vaccine introduction were not truly JE. In such cases, JE vaccine would reduce JE virus infections but do nothing to prevent encephalitis due to other causes.

For vaccine impact to approach the efficacy or effectiveness of a vaccine seen in controlled studies, the quality of vaccine delivery and operational aspects of routine JE immunization must be continuously reviewed and optimized. It is likely that countries that have introduced JE vaccine more recently, which tend to be LICs and LMICs, may benefit from more intensive monitoring and evaluation of vaccine delivery systems to identify areas that can be improved. This comes at a cost and countries should consider whether the amount of persisting JE and the cost to further reduce it is best spent on JE or should be directed to other pressing public health priorities. Funding agencies should also consider that JE vaccine introduction is not a one-and-done SIA. Paying for introduction without committing to a strong routine immunization program and high-quality vaccine delivery is a disservice to the vaccine-introducing country and may keep a vaccine from achieving full impact. Because several of the countries are LICs and LMICs, they may need technical and financial support to improve routine immunization after the initial introduction campaigns.

SIAs have been important adjuncts to the control of measles, rubella, and other highly infectious viruses transmitted person-to-person to eradicate transmission and/or vaccinate hard-to-reach populations but they can be highly disruptive to other public health programs [39]. Interestingly, the countries that introduced JE vaccine prior to 2006 and show the highest impact of JE vaccination used SIAs sparingly during the introduction (S1 and S2 Appendices). Although this study is limited by the high variability in vaccine introduction strategies, making comparisons difficult, several of the most successful JE control programs have occurred in countries where their optimization of routine immunization programs has achieved 80% to 90% vaccine coverage and 90% reduction in JE. Continuous quality improvement of routine immunization programs without disruptive SIAs can achieve rapid reduction in JE and should be considered as a very important JE vaccination strategy. The advantage SIAs offer is very rapid disease reduction, but this needs to be followed with establishment of routine immunization programs with rapid and sustained coverage. SIAs are by themselves a partial solution.

This study has several limitations. The greatest limitation is using multiple sources of data. The WHO-PATH survey data and the biregional data used the same case definitions for probable JE developed in 2002 [23] and both obtained their data from the same ministry/department sources. However, data obtained from the published literature regarding JE incidence, case definitions, type and quality of serologic testing to define probable JE, and methods used to calculate incidence may differ from the WHO-PATH survey and biregional data. However, we used published literature wherein data sources were largely the same country sources as the WHO-PATH survey and Biegional data so we do not expect significant issues with the comparability of estimates due to use of multiple sources. Other limitations include: the possibility of

changes in country strategies for, or quality of, surveillance over time; variability in JE incidence from year to year due largely to variability in conditions for mosquito breeding; variability in consistent data availability which would provide the most accurate estimates of impact; possible lack of direct comparability of incidence rates because countries use different definitions for AE surveillance; and differences in access to and capacity for testing. This means a comprehensive national picture of JE might not be available, and impact estimates may be less accurate if a country is capturing 5% of its cases vs 95%. It is worth noting, however, how well AES can be a marker for JE. Upreti et al. found that JE vaccination in Nepal not only dramatically reduced laboratory confirmed JE, but non-confirmed cases of AES as well, over a five-year follow-up period [40]. Our incidence data for children aged under 15 years in Thailand and Vietnam also affirm this. JE vaccine coverage rates over 85% in Thailand and over 90% in Vietnam have been achieved (S2 Appendix) and a close correlation between post vaccination AE incidence in all age groups and JE incidence in children was demonstrated, suggesting vaccination was responsible for a large portion of measured impact.

Because we compared small regional to large regional/national JE vaccine introduction in countries with limited AES/JE surveillance prior to vaccine introduction, it is likely that we have calculated a lesser impact iof vaccination in Cambodia, Laos, Nepal and especially India where laboratory testing for JE was established just prior to JE vaccine introduction in most districts of Uttar Pradesh. In Chinese Taipei and Japan the initial year of JE vaccine introduction was low coverage and thus designation of the initial year as prevaccination would have had little effect on our calculated impact. Myanmar began JE immunization in mid-November 2017 and so use of 2017 JE surveillance in the pre vaccine period would have had very little effect on impact calculation.

Unlike polio, measles, and rubella, eradication of JE virus will never be within reach due to the persistent circulation of virus within zoonotic reservoirs. Consequently, disease control measures especially vaccination must continue in the countries where good control has already been achieved and may need to be considered in countries and regions where JE virus might become more endemic, e.g., in Pakistan and parts of India and Bangladesh where confirmed cases have not recently been reported but where there is high environmental suitability for *Culex tritaeniorhynchus* [41,42]. In addition we note above recent evidence of JE disease transmission expansion into new countries or previously unaffected regions of countries with established endemic disease [6,27–30].

If JE vaccination is administered in recent and new adopting countries with high integrity of administration as was accomplished in China, Chinese Taipei, Korea and Japan, it is conceivable that the VIMC estimate of a reduction of 100,000 deaths due to JE could be achieved between 2021–2030 [11], assuming the over 90% reduction of JE reported in the early JE vaccine introducers.

Most of the highly JE-endemic countries have already introduced JE vaccine. The further success of vaccine as a JE control measure will undoubtedly be about strengthening the programs in countries where vaccination has already been introduced and continuing and improving disease surveillance to identify new areas of JE virus transmission. What we present here documents a large accomplishment in JE disease reduction and a gradually accumulating knowledge of how to improve and sustain these efforts [43,44].

## Supporting information

**S1 Appendix. Surveillance type, immunization program approach, vaccine used, comparative periods, vaccination coverage, and data sources for countries introducing or**

expanding Japanese encephalitis (JE) vaccination in 2006 or later.
(DOCX)

**S2 Appendix. Surveillance type, immunization program approach, vaccine used, comparative periods, vaccination coverage, and data sources for countries introducing or expanding Japanese encephalitis (JE) vaccination before 2006.**
(DOCX)

**S1 Raw Data. Derived from WHO-PATH protocol and was source of calculations as noted for countries described mostly in S1 but also some subsets of countries described in S2. Other raw data from WHO-PATH BiRegional meetings are available upon request to PATH at cviadatarequests@path.org.**
(PDF)

**S1 Acknowledgements. Membership of the JE Vaccine Global Impact Assessment Team.**
(DOCX)

## Acknowledgments

In alphabetical order by country of association at time of study conduct: Australia-Scott Brown, Queensland Health; Brunei-Linda Lai MOH; Cambodia-Yong Vuttikol, Vichit Ork, (NIP-MOH), Md. Shafikul Hossain WCO Cambodia; India-Prabir Kumar Sen National Vector-Borne Disease Control Program-MOH); Indonesia- Gusti NM Suwarba, Ayu Devi. Udayana University Bali Indonesia, Siwon Choi International Vaccine Institute (Bali, Indonesia); Republic of Korea (South)-Siwon Choi; Laos PDR-Norasingh Sisouveth, Kongxay Phounphenghack (EPI-MOH Laos); Maylasia-Rosa Nani Mudin, Kaihriah Ibrahim MOH; Myanmar-Htar Htar Lin, Aye Mya Chan Thar, Aye Nyein EPI; Papua New Guinea-Berry Ropa MOH; Philippines-Wilda Silva EPI; Singapore-Charlene Tow MOH; Timor Leste-Maria Angela Varel Niha, Manual Mausiri Timor Leste EPI-MOH; Vietnam-Dac Trung Nguyen NIHE, Loan Nguyen EPI.

## Disclaimer

The findings and conclusions of this report are those of the authors and do not necessarily represent the official position of the Centers for Disease Control and Prevention.

## Author Contributions

**Conceptualization:** G. William Letson, Anthony A. Marfin, Jessica Mooney, Susan L. Hills.

**Data curation:** G. William Letson, Jessica Mooney, Huong Vu Minh, Susan L. Hills.

**Formal analysis:** G. William Letson, Anthony A. Marfin, Jessica Mooney, Susan L. Hills.

**Investigation:** G. William Letson, Anthony A. Marfin, Jessica Mooney, Huong Vu Minh, Susan L. Hills.

**Methodology:** G. William Letson, Anthony A. Marfin, Jessica Mooney, Huong Vu Minh, Susan L. Hills.

**Project administration:** G. William Letson, Anthony A. Marfin, Huong Vu Minh.

**Supervision:** G. William Letson, Anthony A. Marfin.

**Validation:** G. William Letson, Anthony A. Marfin, Huong Vu Minh, Susan L. Hills.

**Writing – original draft:** G. William Letson, Anthony A. Marfin, Jessica Mooney, Susan L. Hills.

**Writing – review & editing:** G. William Letson, Anthony A. Marfin, Jessica Mooney, Huong Vu Minh, Susan L. Hills.

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
