## [Decision Letter · Decision Letter 0]

16 Jan 2024

Dear Scientific Advisor for Japanese encephalitis Letson,

Thank you very much for submitting your manuscript "Impact of vaccination against Japanese encephalitis in endemic countries" for consideration at PLOS Neglected Tropical Diseases. As with all papers reviewed by the journal, your manuscript was reviewed by members of the editorial board and by several independent reviewers. In light of the reviews (below this email), we would like to invite the resubmission of a significantly-revised version that takes into account the reviewers' comments. 

We also wish to thank the reviewers for their work especially as this manuscript review coincided with the Christmas vacation.

This work has been expertly and thoroughly considered from immunology, statistics and epidemiology. We request all points raised are addressed, particularly Reviewer 3's methodological concern which include, but are not restricted to the definition of pre-vaccination and coverage. 

We cannot make any decision about publication until we have seen the revised manuscript and your response to the reviewers' comments. Your revised manuscript is also likely to be sent to reviewers for further evaluation.

Sincerely,

Michael W Gaunt, PhD

Academic Editor

Andrea Marzi

Section Editor

Reviewer's Responses to Questions

**Key Review Criteria Required for Acceptance?**

**Methods**

-Are the objectives of the study clearly articulated with a clear testable hypothesis stated?

-Is the study design appropriate to address the stated objectives?

-Is the population clearly described and appropriate for the hypothesis being tested?

-Is the sample size sufficient to ensure adequate power to address the hypothesis being tested?

-Were correct statistical analysis used to support conclusions?

-Are there concerns about ethical or regulatory requirements being met?

Reviewer #1: Objective is clear, study design is appropriate.

All parts of methodology have been addressed clearly including population description and the statistical anlaysis.

No concerns over the ethical or regulatory requirements.

Reviewer #2: yes

Reviewer #3: I believe that the first three bullets in the methods section are meant to represent an overview of the methods that follow but this is not clear as written, particularly since the bullets don’t include the same text that is used for the section headings that follow. Consider adding/editing language to clarify for the reader. Also, these sections include some duplication of information (in main text and bullets) and would be useful to streamline.

Methods for Thailand and Vietnam need not be in parentheses as these are important details for the reader.

The methods for determining vaccine coverage don’t correspond to the data collected through the survey. For example, the survey collected data on the number of doses of vaccine given, but the coverage suggests a list of all children along with their vaccination status. Where did those data come from? Other population-based surveys?

The statement following statement would benefit from additional clarifications – were these the definitions of a confirmed case for countries? Perhaps this point is better addressed in the discussion rather than the methods? “Because most of the reported JE incidence data are based on the serologic results of serum IgM ELISA, [23] the impact results reported here are considered to be the reduction of probable JE cases.”

I presume that the calculations were made for all age groups or for those under the age of 15 depending on the data available from the countries, correct?

The following statement is not clear – were these data for Sarawak or Malaysia nationwide? “…and from Sarawak (Malaysia), data represent impact from introduction primarily in Sarawak state and in Maylasia nationwide.”

Additional details for these methods would be useful to include: “We present data for u15 JE incidence in the after-vaccine introduction period to validate use of AE/viral encephalitis reduction in these countries as a estimator of JE as a principal cause of AE/viral encephalitis in these countries.”

The following methods are not quite clear. Can authors add more details? Why was 90% chosen? Also, Figure 1 does not appear to present a rolling average of all the countries studied.

“To determine the number of years for vaccine introduction to achieve a sustained of 90% or greater

reduction in JE incidence, the mean annual u15 or all-age JE incidence was determined for the four years

prior to the start of national vaccine introduction. A rolling 4-year average of all-age JE incidence was

calculated until the first 4-year period where a 90% or greater reduction in incidence was identified

(Figure 1). The number of years from the time of JE vaccine introduction to the first year of a 4-year

period with sustained reduction was noted.”

**Results**

-Does the analysis presented match the analysis plan?

-Are the results clearly and completely presented?

-Are the figures (Tables, Images) of sufficient quality for clarity?

Reviewer #1: Results are clear, figures and tables are sufficient quality for clarity.

Reviewer #2: yes

Reviewer #3: Table 1 – The pre-introduction study years frequently include the year that the vaccine was introduced, and in some instances, includes many years after the introduction. It’s unclear why these years are termed ‘pre-introduction’. Certainly, this could explain why the incidence change is low? For ROK and Malaysia the time periods are distinct.

In the titles of tables, please include details about the age groups and countries that are represented to ensure the table can stand alone.

For the measured decreases in incidence many years after the introduction of vaccine (Thailand and Vietnam) the ability to infer declines as solely the result of vaccination is greatly reduced. Other social and environmental changes could also be relevant, particularly given the rapid social and ecological changes in these countries over these time periods.

For the countries that did achieve 90% reductions, were other differences also part of why it took many years for JE incidence to decline? I also presume that the type of vaccination campaign would have a large impact on timing – for example, if the campaign only vaccinated infants, it would take 15 years for the highest risk group to all be covered. This additional context seems important to highlight. 

Overall, making the link between impact, especially over longer time periods, would be bolstered by showing the increases in vaccination coverage over time to explain trends.

**Conclusions**

-Are the conclusions supported by the data presented?

-Are the limitations of analysis clearly described?

-Do the authors discuss how these data can be helpful to advance our understanding of the topic under study?

-Is public health relevance addressed?

Reviewer #1: Conclusion is appropriate given the limitation described.

Reviewer #2: yes

Reviewer #3: The emphasis on pre versus post 2006 seems a bit strange. What is the rationale for this? It’s clear that time since introduction is important to note (as is the target of the vaccine campaigns – see above – but additional rationale for why 2006 was a major turning point would help the reader.

As noted above, could ecological context explain some changes that occurred over 30 years? For example, increased urbanization and contact with vectors? 

The text refers to some of the countries being low-income countries, but I think they’re all at least lower middle income, no? I suppose there is a difference between the country’s economic status when the introduced vaccination and the status today but all references about economic categories report one single category. Perhaps worth discussing any changes over time that might be relevant to the story.

What is the evidence supporting the claim that the virus might become more highly endemic in some Asian countries? Would be good to cite support.

**Editorial and Data Presentation Modifications?**

Reviewer #1: SIAs was not spelled out in the discussion session in the first instance.

Reviewer #2: Table 1 – suggest putting the info in the superscript “2” in the table – ie “incidence/100,000 under 15 population”

Reviewer #3: Minor issues:

Check for typos – no line numbers made it difficult to mention specific instances.

“u15” is not a standard acronym so suggest avoiding.

Best to avoid sentence construction using “respectively” as it requires the reader to go back and re-read the sentence to understand what is being said. 

The abstract would benefit from some edits to clarify a few key issues:

“…in children younger than 15-years-old and in persons of any age…” Can shorten to just say persons of any age since this will cover the range.

Can authors clarify how many endemic countries there are in the abstract? Only 13 or is this a subset? (Methods section methods 23?)

What were the three data collection methods used in the study?

How long was impact measured following vaccination? In Thailand and Vietnam, what were the reductions? Please provide the data to support the main conclusions.

What would there be a difference in impact for countries that introduced before and after 2006? This is unclear.

**Summary and General Comments**

Reviewer #1: (No Response)

Reviewer #2: Letson et al have examined the effects of the introduction of JE vaccination programs in 14 countries. This has been done using surveillance data rather than modelled, which is intended to give greater accuracy. The paper is well written and provides a compelling case to policy makers for JE vaccination programs. Overall I have very few comments and I recommend publication of the work.

The authors point out that an estimated 100,000 deaths could be averted by JE vaccination between 2021 and 2030. Could they add a comment to the manuscript on whether their own findings support or refute this?

I am not an epidemiologist, so I am not qualified to comment on the specific methods. The authors have pointed out the major weakness being the variable source of the data. They have not elaborated much on what effect that might have had on their conclusions, if any. I suggest adding a comment on this to the discussion. (I suspect that this does not impact the overall significance of this paper very much.)

Minor points:

Page 9 last para – suggest starting with Under 15 JE incidence

Table 1 – suggest putting the info in the superscript “2” in the table – ie “incidence/100,000 under 15 population”

Please note I am not certain about the data availability. I am sure this can be easily resolved.

Reviewer #3: JE vaccines are a powerful public health tool and the authors' efforts to identify the magnitude of their impact on disease is laudable. These results could be used by the countries included to better understand how their program compares to others in the region, and the results could be used by other countries that have not yet introduced vaccines. Readers would be interested in this paper.

However, as currently written, there are some methodological concerns with how the pre-vaccine periods are defined and some details lacking about vaccine coverage, particularly over time. The claims made about the relationship between vaccines and encephalitis incidence could be greatly strengthened by addressing these concerns.

PLOS authors have the option to publish the peer review history of their article (what does this mean?). If published, this will include your full peer review and any attached files.

Reviewer #1: Yes: Gary Low

Reviewer #2: No

Reviewer #3: Yes: Emily S Gurley
---

## [Decision Letter · Decision Letter 1]

2 May 2024

Dear Dr. Letson,

Thank you very much for submitting your manuscript "Impact of vaccination against Japanese encephalitis in endemic countries" for consideration at PLOS Neglected Tropical Diseases. As with all papers reviewed by the journal, your manuscript was reviewed by members of the editorial board and by several independent reviewers. The reviewers appreciated the attention to an important topic.

Based on the reviews, we are likely to accept this manuscript for publication, providing that you modify the manuscript according to the review recommendations described by Reviewer 3.

Sincerely,

Michael W Gaunt, PhD

Academic Editor

Andrea Marzi

Section Editor

# Comments from AE

Review 3 is requesting minor revisions, whilst Reviewers 1 and 2 have accepted the ms. The outstanding revisions are relatively simply but there are moderate numbers of changes required to the ms therefore a "minor revision" status is activated. I intend to personally check compliance with each request on resubmission and thereon would hope to anticipate a seamless movement to acceptance.

Reviewer's Responses to Questions

**Key Review Criteria Required for Acceptance?**

**Methods**

-Are the objectives of the study clearly articulated with a clear testable hypothesis stated?

-Is the study design appropriate to address the stated objectives?

-Is the population clearly described and appropriate for the hypothesis being tested?

-Is the sample size sufficient to ensure adequate power to address the hypothesis being tested?

-Were correct statistical analysis used to support conclusions?

-Are there concerns about ethical or regulatory requirements being met?

Reviewer #1: Yes

Reviewer #2: NA

Reviewer #3: On page 9, “Because a large portion of AE and viral encephalitis is due to JE in children in these countries” Can the authors provide the % here? The definition of ‘large’ could vary widely. Also, the authors should consider moving this paragraph to the end of the methods so that they can present the methods for the JE specific data first, then talk about the non-JE data analysis.

**Results**

-Does the analysis presented match the analysis plan?

-Are the results clearly and completely presented?

-Are the figures (Tables, Images) of sufficient quality for clarity?

Reviewer #1: yes

Reviewer #2: NA

Reviewer #3: Table 1 – pre-introduction years are sometimes after the year of introduction which needs additional context to make clear. Perhaps a note in the text and footnote on the table so that the table will not be interpreted as being inconsistent. In the methods, the authors say that they defined ‘significant program expansion’ to determine which countries were included in the analysis but adding how that was done in the text would be useful. Were the campaigns in the table one-time activities? How could coverage be >100% Might be useful to mention that in the limitations.

**Conclusions**

-Are the conclusions supported by the data presented?

-Are the limitations of analysis clearly described?

-Do the authors discuss how these data can be helpful to advance our understanding of the topic under study?

-Is public health relevance addressed?

Reviewer #1: yes

Reviewer #2: NA

Reviewer #3: Discussion – the authors use the example of increases in dengue transmission in many areas of the time period of JE vaccine introduction to argue that the JE vaccine is primarily responsible for the declines in JE incidence over time, rather than other changes to the environment. However, dengue mosquito vectors (Aedes aegypti in particular) thrive in urban settings, which have growing quickly over the time period of this study. In contrast, Culex species (a major vector for JE) prefer rural environments, which may have been declining as urban areas have developed. I appreciate the attempt to address this comment, but I don’t find the dengue example particularly convincing. The Zika and Chikungunya arguments don’t add anything, in my opinion, also because they are related to different mosquito species with different habitat preferences. 

Lines 396-397: What is the evidence that JE might become more endemic in these countries? A reference would be useful here.

The authors note in the introduction that modeled estimates of vaccine impact have been made but that they have not bee validated with epidemiologic data. It would be useful in the discussion to provide some analysis of how the modeled estimates compare to these results.

**Editorial and Data Presentation Modifications?**

Reviewer #1: The SIA acronym appeared first in 3rd paragraph of discussion section, line 314. The author did explained the acronym but it was at the second appearance, at 6th paragraph, line 342.

Reviewer #2: NA

Reviewer #3: In the author summary, the year 2003 is mentioned as the year when efforts began to support JE vaccine programs, but other places in the paper refer to 2006.

Abstract:

Methodology/principal findings – For the decreases in incidence, it would be useful to give the year at which these reductions were identified. Are the data through 2023? Or might it be better to report how many years after introduction these reductions were realized? The last sentence of this section is difficult to interpret – please include some indication in the abstract about why these results are reported differently.

Minor edits:

Bottom of page 18 – ‘vatiability’ should be ‘variability’.

Line 390 – there is a stray quotation mark in the text.

I appreciate that the authors have addressed the comment about the u15 acronym. However, the text is still a bit unclear in some places where the term ‘under 15’ is used and could benefit from a review.

**Summary and General Comments**

Reviewer #1: (No Response)

Reviewer #2: The authors have satisfactorily responded to all the points. I do tend to agree with the other reviewer who asked to change “u15.” However, the changes have resulted in some slightly clunky phrases. Eg:

Line 189 “all under 15 year children” would be better phrased as “all children under 15.”

“Under 15 years JE incidence” which is used frequently would be better phrased as “JE incidence in children under 15.”

Whilst these are small points, it would seem worth changing them, which should not be a big job.

Line 387 – typo “iof”

With these changes the article could be accepted without further review.

Reviewer #3: The authors did a thorough job in addressing the reviewers' concerns. Only a few additional points for their consideration are included in this review.

PLOS authors have the option to publish the peer review history of their article (what does this mean?). If published, this will include your full peer review and any attached files.

Reviewer #1: Yes: Gary Low

Reviewer #2: No

Reviewer #3: No

Figure Files:

Data Requirements:

Reproducibility:

References

---

## [Editor Report · Decision Letter 2]

19 Jul 2024

Dear Dr. Letson,

We are pleased to inform you that your manuscript 'Impact of vaccination against Japanese encephalitis in endemic countries' has been provisionally accepted for publication in PLOS Neglected Tropical Diseases.

Best regards,

Michael W Gaunt, PhD

Academic Editor

Andrea Marzi

Section Editor

---

## [Editor Report · Acceptance letter]

28 Aug 2024

Dear Dr. Letson,

We are delighted to inform you that your manuscript, "Impact of vaccination against Japanese encephalitis in endemic countries," has been formally accepted for publication in PLOS Neglected Tropical Diseases.

Best regards,

Shaden Kamhawi

co-Editor-in-Chief

Paul Brindley

co-Editor-in-Chief
